# Arginine Methylation of hnRNPK Inhibits the DDX3-hnRNPK Interaction to Play an Anti-Apoptosis Role in Osteosarcoma Cells

**DOI:** 10.3390/ijms22189764

**Published:** 2021-09-09

**Authors:** Chiao-Che Chen, Jen-Hao Yang, Shu-Ling Fu, Wey-Jinq Lin, Chao-Hsiung Lin

**Affiliations:** 1Department of Life Sciences and Institute of Genome Sciences, National Yang Ming Chiao Tung University, Taipei 112, Taiwan; jj30108018.y@nycu.edu.tw (C.-C.C.); tim6260240@gmail.com (J.-H.Y.); 2Institute of Traditional Medicine, National Yang-Ming Chiao Tung University, Taipei 112, Taiwan; slfu@ym.edu.tw; 3Institute of Biopharmaceutical Sciences, National Yang Ming Chiao Tung University, Taipei 112, Taiwan; wjlin@ym.edu.tw; 4Proteomics Research Center, National Yang Ming Chiao Tung University, Taipei 112, Taiwan

**Keywords:** apoptosis, hnRNPK, DDX3, DNA damage, protein–protein interaction

## Abstract

Heterogeneous nuclear ribonucleoprotein K (hnRNPK) is an RNA/DNA binding protein involved in diverse cell processes; it is also a p53 coregulator that initiates apoptosis under DNA damage conditions. However, the upregulation of hnRNPK is correlated with cancer transformation, progression, and migration, whereas the regulatory role of hnRNPK in cancer malignancy remains unclear. We previously showed that arginine methylation of hnRNPK attenuated the apoptosis of U2OS osteosarcoma cells under DNA damage conditions, whereas the replacement of endogenous hnRNPK with a methylation-defective mutant inversely enhanced apoptosis. The present study further revealed that an RNA helicase, DDX3, whose C-terminus preferentially binds to the unmethylated hnRNPK and could promote such apoptotic enhancement. Moreover, C-terminus-truncated DDX3 induced significantly less apoptosis than full-length DDX3. Notably, we also identified a small molecule that docks at the ATP-binding site of DDX3, promotes the DDX3-hnRNPK interaction, and induces further apoptosis. Overall, we have shown that the arginine methylation of hnRNPK suppresses the apoptosis of U2OS cells via interfering with DDX3–hnRNPK interaction. On the other hand, DDX3–hnRNPK interaction with a proapoptotic role may serve as a target for promoting apoptosis in osteosarcoma cells.

## 1. Introduction

Apoptosis is a form of programmed cell death that occurs in response to different cellular stimuli, including oxidative stress, endoplasmic reticulum (ER) stress, and DNA damage. In addition, PKCδ, a serine/threonine kinase, is critical in DNA damage-induced apoptosis [1] and plays a proapoptotic role through the phosphorylation of essential targets such as tumor suppressor p53 and the cell cycle checkpoint protein Rad9 [2,3]. PKCδ also mediates the serine 302 (Ser302) phosphorylation of heterogeneous nuclear ribonucleoprotein K (hnRNPK) and triggers apoptosis induction [4,5]. HnRNPK participates in diverse cellular processes, such as chromatin remodeling, transcription, RNA splicing, mRNA stabilization, translation, and the DNA damage response, through direct interaction with DNA, RNA, or proteins [6,7]. Notably, a supporting role of hnRNPK in cancer progression has been implied due to its elevated expression in various cancers and its positive association with aggressive metastasis, poor prognosis, and a cancer stem cell phenotype [6,8,9,10,11,12,13]. Alternatively, hnRNPK is also recognized as a transcriptional coactivator of p53 that initiates cell cycle arrest and apoptosis upon DNA damage [14]. The detailed mechanism regarding how hnRNPK switches between its apoptosis-promoting and suppressive roles in cancer cells remains unclear.

We previously demonstrated that the PKCδ-mediated Ser302 phosphorylation of hnRNPK is also important in the DNA damage-induced apoptosis of U2OS osteosarcoma cells, whereas such phosphorylation is negatively regulated by nearby methylations at arginines 296 and 299 (Arg296/299), which are located in the protein-interacting region of hnRNPK [15]. Notably, methylation-defective hnRNPK (hnRNPK^MD^), which is generated by the removal of the arginine methylation sites through R296K/R299K mutation, exhibits a significant increase in Ser302 phosphorylation and promotes the apoptosis of U2OS cells [15]. The K homology (KH) domains of hnRNPK are essential for nucleotide binding, while the K-interactive (KI) region is responsible for the diverse protein–protein interactions of hnRNPK [7]. The functional importance of this KI region has been implied by numerous studies on hnRNPK’s protein–protein interactions, which are involved in breast cancer progression [8], glioma cell migration [9], cell cycle arrest and apoptosis [14], hepatitis C virus production [16], and osteoclast differentiation [17]. Moreover, arginine methylation in the KI region of hnRNPK has been shown to reduce its binding with the SH3 domain of c-Src [18]. Therefore, it is suggested that Arg296/299 methylation and Ser302 phosphorylation in the KI region of hnRNPK may regulate DNA damage-induced apoptosis through protein–protein interactions.

Several studies have implied that DDX3 and hnRNPK are present in the same protein complex [19,20], and DDX3 is also known as an RNA helicase involved in tumorigenesis and apoptosis [21,22,23,24]. In addition, the coimmunoprecipitation of hnRNPK and DDX3 has been reported in the context of the apoptosis of pancreatic β cells [25]. In the present study, we showed that DDX3 is critical to the apoptosis enhancement mediated by hnRNPK^MD^. Moreover, we further demonstrated that hnRNPK–DDX3 interaction could play an apoptosis-promoting role in U2OS cells under DNA damage conditions. Our discovery of the potential crosstalk between the DDX3 and hnRNPK cascades may provide a new strategy for apoptosis induction in cancer cells.

## 2. Materials and Methods

### 2.1. Plasmids and Antibodies

WT pET23a-Trx-hnRNPK was prepared according to a previous study [26]. The cDNA segment of the KI region (hnRNPK amino acid sequence: 240–336) was subcloned into the expression vector pET23a-Trx by KpnI/XhoI digestion. Full-length cDNA DDX3 and its truncated variants were inserted into the PGEX-5X-1 vector. In addition, full-length cDNAs encoding human DDX3 and C-terminal truncated DDX3 were inserted into a pCDNA4-myc/His vector. The cDNA fragment of hnRNPK or hnRNPK-ΔKI was subcloned into the expression vector of pCDNA3-N-terminal Venus through KpnI/EcoRI digestion.

The antibodies used in the present study were purchased from vendors as indicated: Anti-DDX3 (#54169) from Arigo Biolaboratories (Hsinchu, Taiwan); anti-JUND (#102135) and anti-GAPDH (#100118) were from GeneTex lnc (San Antonio, TX, USA); Anti-β-actin (AC-15), anti-hnRNPK/J (3C2), and anti-Flag (M2) were from Sigma-Aldrich (St Louis, MO, USA); and anti-Myc tag (#9B11) and anti-cleaved caspase-3 (#9661) were from Cell Signaling Technology (Beverly, MA, USA).

### 2.2. Preparation of His_6_-Tagged Proteins

*Escherichia coli* BL21 DE3 pLysS cells were transformed with either pET23a-Trx-hnRNPK or pET23a-Trx-KI and were then grown in LB medium. Production of the recombinant His_6_-tagged proteins was induced by the addition of 0.4 mM isopropyl β-d-1-thiogalactopyranoside (IPTG) to LB medium when the optical density at a wavelength of 600 nm (OD_600_) reached 0.4–0.6. After 3 h of induction at 30 °C, the cells were harvested, resuspended in binding buffer (5 mM imidazole, 0.5 M NaCl, and 20 mM Tris-HCl pH 8), and underwent sonication. The cell lysates were subsequently incubated with Ni-nitrilotriacetic acid (NTA) agarose beads (Qiagen, Chatsworth, CA, USA) using the procedure described by the manufacturer. The recombinant His_6_-tagged proteins were eluted with imidazole and were purified by dialysis to remove the salt and imidazole.

### 2.3. Preparation of GST-Tagged Fusion Proteins

*Escherichia coli* BL21 DE3 pLysS cells were transformed with pGEX-5X-1 plasmids and were then grown in LB medium. Production of the recombinant GST-tagged proteins was induced by the addition of 0.2 mM IPTG to LB medium when the OD_600_ reached 0.4–0.6. After 3 h of induction at 30 °C, the cell pellets were harvested, washed with cold PBS, and resuspended in binding buffer (PBS containing 5% glycerol, 0.5% Triton X-100, 1 mM EDTA, 1 mM EGTA, 1 mM DTT, and protease inhibitors). The cell lysates were prepared through the sonication of the cell pellets and were subsequently incubated with glutathione-Sepharose 4 Fast Flow beads (Amersham Biosciences, Uppsala, Sweden) using the procedure described by the manufacturer.

### 2.4. Pull-Down Assay Using GST-Tagged Proteins

U2OS cells were lysed, and cell extracts were incubated with bead-bound GST-tagged proteins in binding buffer (PBS containing 0.5% NP-40, 0.25% sodium deoxycholate, 5% glycerol, and protease inhibitors) at 4 °C for 1 h. After three washes with binding buffer, the bead-bound proteins were released through boiling in sample buffer and were directly analyzed by means of SDS-PAGE. For the pull-down assay of the GST-DDX3 variants, recombinant hnRNPK or the KI segment was used for the binding assay instead.

### 2.5. Cell Culture and Transfection

U2OS-WT and U2OS-2RK cells were cultured with Dulbecco’s modified Eagle’s medium (Gibco BRL, Grand Island, NY, USA) containing 10% fetal bovine serum (HyClone, Logan, UT, USA), L-glutamine, penicillin, and streptomycin (Gibco BRL, Grand Island, NY, USA). The cells were grown in a humidified 5% CO_2_ incubator at 37 °C. Subsequent transient transfections with different DDX3-WT or -mutant proteins were performed using TurboFect™ (Fermentas, Carlsbad, CA, USA) according to the manufacturer’s instructions.

### 2.6. Immunoprecipitation and Western Blot Analysis

Immunoprecipitation and Western blot analyses were performed as previously described [15]. Briefly, the cells were harvested and lysed in PBS buffer containing 0.5% Triton X-100 and protease inhibitor cocktail (Sigma-Aldrich). The resulting cell lysate was incubated with anti-flag primary antibodies and protein G Sepharose beads (GE Healthcare Bio-Sciences, Uppsala, Sweden) at 4 °C for 3 h. After centrifugation and the removal of the supernatant, the protein-bound beads were washed three times with PBS buffer containing 0.5% Triton X-100 followed by boiling at 95 °C in sample buffer to release the bound proteins. Protein samples were then separated by SDS-PAGE and were subsequently transferred to a PVDF membrane (Millipore, Darmstadt, Germany). The membranes were incubated with a blocking solution comprised of 5% skimmed milk and TBST buffer (0.05% Tween-20) for 1 h. After further incubation with primary antibody at 4 °C overnight, the membranes were washed with TBST buffer three times followed by incubation with horseradish peroxidase-conjugated secondary antibodies for 1 h. The antibody signals reacted with ECL Western blotting Detection Reagent (Millipore, Darmstadt, Germany) and were detected by exposing the membrane to X-ray film.

### 2.7. RNA Interference

SMARTpool siRNA containing four siRNAs targeting human DDX3 was purchased from Sigma-Aldrich. The specific sequences are shown here:5′-GAUUCACUGACCUUAGUGU[dT][dT]-3′;5′-GCAAAUACUUGGUGUUAGA[dT][dT]-3′;5′-GAAACAGUCGCUGGUGUGA[dT][dT]-3′;5′-GAAAUACGAUGACAUUCCA[dT][dT]-3′.

Knockdown of DDX3 was achieved by transfection with the siRNA at a final concentration of 25 nM using TurboFect (Fermentas, Carlsbad, CA, USA), and the resulting cells were analyzed 36 h later.

### 2.8. TUNEL Assay

U2OS-2RK cells were transfected with DDX3-myc, siRNAs for DDX3 knockdown, DDX3^DQAD^, or C-terminal truncated DDX3 for 24 h. Transfected cells were subsequently incubated with etoposide only (50 μM), etoposide together with RK-33 (5 μM), AMPPNP (5 μM), or AMP (5 μM) for the indicated times. The cells were harvested and were fixed with 4% formaldehyde. DNA double-strand breaks of apoptotic cancer cells were labeled by using the In Situ Cell Death Detection Kit (Roche Applied Science). The percentage of labeled apoptotic cells was detected by flow cytometry.

### 2.9. Statistical Analysis

The bars represent the mean ± SD as shown in the figures. Comparisons between two groups were performed using a two-tailed Student’s *t* test. When analyzing differences between different groups, *p <* 0.05 was considered significant. All the of the comparison results are indicated with the level of significance (* as *p <* 0.05; ** as *p <* 0.01; *** as *p <* 0.001).

## 3. Results

### 3.1. DDX3 Is Critical to the Apoptosis Enhancement Mediated by hnRNPK^MD^

We first showed that DDX3 indeed directly interacts with either full-length hnRNPK or its KI region in vitro (Figure 1A). Previously, the U2OS-WT and U2OS-2RK cell lines were established through the replacement of endogenous hnRNPK with either Flag-tagged wild-type (WT) hnRNPK or hnRNPK^MD^, respectively, in U2OS cells [15]. Upon DNA damage, the U2OS-2RK cells exhibited higher levels of apoptosis than the U2OS-WT cells (Appendix A). Whether the DDX3–hnRNPK interaction is involved in the apoptosis regulation of U2OS-WT and U2OS-2RK cells was further investigated. As shown in Figure 1B, hnRNPK^MD^ was able to precipitate more DDX3 than WT hnRNPK, suggesting that DDX3 interacts more strongly with hnRNPK^MD^ than with hnRNPK. In addition, the hnRNPK^MD^ in U2OS-2RK cells exhibited a much higher level of DDX3 interaction under DNA damage conditions than the wild-type hnRNPK in U2OS-WT cells (Figure 1C). Because hnRNPK^MD^ exhibits higher Ser302 phosphorylation, which is essential for DNA damage-induced apoptosis [15], we examined whether the hnRNPK–DDX3 interaction is affected by Ser302 phosphorylation. As shown in Figure 1D, the loss of Ser302 phosphorylation in hnRNPK^MD^ (R296K/R299K, S302A) significantly reduced the DDX3–hnRNPK^MD^ interaction under DNA damage conditions. These results suggest that DDX3 preferentially binds to hnRNPK^MD^ in the presence of Ser302 phosphorylation and under DNA damage conditions.

To determine whether DDX3 is involved in the apoptosis enhancement mediated by hnRNPK^MD^ in U2OS-2RK cells, DDX3 knockdown and overexpression were performed to evaluate the influences on apoptosis upon DNA damage. While U2OS-2RK cells exhibited more cleaved caspase-3 than U2OS-WT cells upon etoposide treatment, the knockdown of DDX3 by small interfering RNA (siRNA) in U2OS-2RK cells significantly attenuated the etoposide-induced elevation of cleaved caspase-3 (Figure 2A). Similarly, the knockdown of DDX3 in U2OS-2RK cells also reduced the apoptosis enhancement mediated by hnRNPK^MD^ upon DNA damage (Figure 2B). Alternatively, the overexpression of DDX3 in U2OS-2RK cells significantly increased the level of cleaved caspase-3 and apoptosis upon DNA damage (Figure 2C,D). Taken together, our results suggest that DDX3 is critical to the apoptosis enhancement mediated by hnRNPK^MD^ in U2OS-2RK cells under DNA damage conditions.

### 3.2. The C-Terminal Region of DDX3 Is Responsible for the hnRNPK–DDX3 Interaction

While hnRNPK interacts with DDX3 through its KI region (Figure 1A), the domain responsible for the hnRNPK interaction with DDX3 remains unknown. To further illustrate the role of DDX3 in the hnRNPK-dependent regulation of apoptosis upon DNA damage, we next investigated the hnRNPK-interacting region of DDX3. The GST-tagged full-length (GST-DDX3^1–662^), N-terminal region (GST-DDX3^1–226^), catalytic core (GST-DDX3^227–534^), and C-terminal region (GST-DDX3^535–662^) of DDX3 were generated and were incubated with recombinant His_6_-tagged hnRNPK (Figure 3A). As shown in Figure 3B, the C-terminal region (DDX3^535–662^) of DDX3 exhibited the strongest interaction with His-hnRNPK. Similarly, the C-terminal region of DDX3 also interacted with the His_6_-tagged KI region (Figure 3C). Moreover, the lysates of U2OS-WT and U2OS-2RK cells were used in pull-down assays to determine whether WT hnRNPK and hnRNPK^MD^ exhibit different interactions with the C-terminal region of DDX3. As shown in Figure 3D, hnRNPK^MD^ exhibited a stronger interaction with the DDX3 C-terminus than WT hnRNPK did. These results indicate that hnRNPK preferentially interacts with the C-terminal region of DDX3, whereas the absence of arginine methylation in hnRNPK further favors this interaction.

### 3.3. C-Terminus-Truncated DDX3 Could Not Support the Apoptosis Enhancement Mediated by hnRNPK^MD^ upon DNA Damage

We next investigated whether C-terminus-truncated DDX3 sustains its binding with hnRNPK^MD^ as well as its ability to induce apoptosis upon DNA damage. Our results showed that the C-terminal truncation of DDX3 abolished its protein interaction with hnRNPK^MD^ (Figure 4A). In addition, compared to the U2OS-2RK cells that overexpressed full-length DDX3, the U2OS-2RK cells that overexpressed DDX3 truncated at the C-terminus did not induce comparable levels of apoptosis and caspase-3 cleavage (Figure 4B,C). Taken together, these results suggest that the interaction between hnRNPK and the C-terminus of DDX3 is closely associated with the apoptosis enhancement mediated by hnRNPK^MD^ upon DNA damage.

### 3.4. Mutation of the DEAD Motif in DDX3 Reduces Its hnRNPK Interaction and Impairs Its Apoptosis-Promoting Ability in U2OS Cells under DNA Damage

DDX3 is an RNA helicase that is involved in cancer development and apoptosis [27], whereas whether the helicase activity of DDX3 affects its interaction with hnRNPK remains unknown. It has been shown that a glutamate-to-glutamine mutation in the DEAD motif of DDX3 (DDX3^DQAD^) impairs its ATPase and RNA helicase activity [28,29]. Therefore, we next determined whether DDX3^DQAD^ affects the apoptosis enhancement by hnRNPK^MD^ using TUNEL assay and Western blot analysis. As shown in Figure 5A,B, overexpression of the DDX3^DQAD^ in U2OS-2RK cells did not induce comparable levels of apoptosis and caspase-3 cleavage to those of the U2OS-2RK cells that were overexpressed with wild-type DDX3, indicating that the DEAD motif of DDX3 is important for the hnRNPK-dependent regulation of apoptosis. We next examined whether DDX3^DQAD^ affects the interaction between DDX3 and hnRNPK^MD^, and co-immunoprecipitation experiments of the hnRNPK^MD^ with either wild-type DDX3 or DDX3^DQAD^ were performed in the U2OS-2RK cells. As shown in Figure 5C, hnRNPK^MD^ exhibits significantly stronger interaction with wild-type DDX3 than with DDX3^DQAD^. Taken together, our results demonstrate that the DEAD motif in DDX3 is important for hnRNPK–DDX3 interaction as well as for apoptosis enhancement by hnRNPK^MD^.

### 3.5. RK-33 Promotes the Apoptosis Enhancement Mediated by hnRNPK^MD^ by Increasing the hnRNPK–DDX3 Interaction

Because the helicase-inactive DDX3^DQAD^ exhibited less hnRNPK interaction than wild type DDX3, we next examined whether the inhibition of DDX3 helicase activity by small molecules also affects DDX3–hnRNPK interaction. The DDX3 helicase inhibitor RK-33 is a ring-expanded nucleoside analog that can dock at the ATP binding site and can specifically inhibit the helicase activity of DDX3 [30,31], whereas RK-33 does not affect closely related helicases such as DDX5 and DDX17 [30]. We therefore conducted a coimmunoprecipitation experiment with hnRNPK and DDX3 in U2OS-2RK cells with or without RK-33 treatment. As shown in Figure 6A, hnRNPK^MD^ was able to precipitate more DDX3 in U2OS-2RK cells treated with RK-33 than in control cells. Moreover, RK-33 significantly promoted the DDX3–hnRNPK interaction under DNA damage conditions (Figure 6B). Accordingly, we further examined whether RK-33 treatment could affect the apoptosis of U2OS-2RK cells under DNA damage conditions. As shown in Figure 6C, the percentage of apoptotic cells was significantly increased in U2OS-2RK cells treated with both RK-33 and etoposide. In addition, RK-33 also increased the cleaved caspase-3 levels of U2OS-2RK cells under DNA damage conditions (Figure 6D). On the other hand, a subunit of the AP-1 transcription factor JUND has been reported to participate in apoptosis regulation, which involves hnRNPK and DDX3 [25,32,33]. We thus further determined whether JUND mediates the apoptosis enhancement promoted by DDX3–hnRNPK^MD^ interaction upon DNA damage. As shown in Figure 6E, the JUND level was slightly increased in U2OS-2RK cells treated with etoposide but was significantly elevated upon cotreatment with etoposide and RK-33. In addition, such elevation of JUND was not observed in the U2OS-WT cells under DNA damage conditions (Appendix A). Taken together, these results suggest that RK-33 promotes hnRNPK–DDX3 interaction to significantly increase the level of hnRNPK-dependent apoptosis, which leads to upregulation of JUND in the downstream cascade.

### 3.6. Other ATP Site Binding Ligands of DDX3 Did Not Promote hnRNPK^MD^-Mediated Apoptosis Enhancement

We further investigated whether another DDX3 inhibitor, avenanthramide A (AVN A), had the same effect as RK-33 on DNA damage-induced apoptosis and hnRNPK–DDX3 interaction. AVN A also docks at the ATP binding cleft and reduces the ATPase activity of DDX3 [32]. As shown in Appendix A, treatment with AVN A did not increase cell apoptosis or the hnRNPK–DDX3 interaction in U2OS-2RK cells under DNA damage conditions. Because RK-33 and AVN A target different residues of DDX3 [30,32], we further examined whether AMP or the ATP analog AMPPNP regulates DDX3–hnRNPK interaction through docking at the ATP binding site of DDX3. However, neither AMPPNP nor AMP promoted hnRNPK–DDX3 interaction (Figure 7A). Similarly, AMPPNP and AMP did not increase the percentage of apoptotic cells or the level of cleaved caspase-3 in U2OS-2RK cells under DNA damage conditions (Figure 7B,C). The crystal structures of AMPPNP-bound DDX3 and AMP-bound DDX3 as well as the apo form of DDX have been reported [33]. As shown in Figure 7D, the N-terminal alignment of these three structures revealed the distinct spatial arrangement of their C-terminal tails. Because we have shown that the C-terminal region of DDX3 is responsible for hnRNPK interaction, the docking of RK-33 in DDX3 may place its C-terminus in an orientation that favors the interaction of DDX3 with hnRNPK.

## 4. Discussion

The Arg296/299 methylation of hnRNPK has been previously reported to suppress the PKCδ-mediated Ser302 phosphorylation of hnRNPK as well as the apoptosis of U2OS cells [15]. To illustrate the biochemical details regarding how the arginine methylation of hnRNPK regulates PKCδ-dependent apoptosis, we herein demonstrated that DDX3 interacts with hnRNPK in a methylation-interfering manner (Figure 1B,C). Alternatively, Ser302 phosphorylation is also critical to this DDX3–hnRNPK^MD^ interaction, which is supported by the loss of apoptosis enhancement upon S302A mutation in hnRNPK^MD^ (Figure 1D). We concluded that the absence of arginine methylation in hnRNPK^MD^ not only promotes PKCδ-mediated Ser302 phosphorylation but also increases DDX3–hnRNPK^MD^ interaction, which leads to the enhancement of PKCδ-dependent apoptosis upon DNA damage.

DDX3 is biochemically characterized by the conserved catalytic core, which contains 12 conserved motifs and exhibits both ATPase and RNA duplex-unwinding activities [34]. In addition, this catalytic core is flanked by unique N- and C-terminal sequences that are distinct from those of other DDX proteins. Moreover, DDX3 has been reported to participate in many cellular events, including cell cycle regulation, cellular differentiation, cell survival, and apoptosis [27,35]. Furthermore, the protein–protein interactions of DDX3 have been shown to be involved in regulating cellular processes, for example, the interaction of its C-terminus with EIF4E in the translation suppression or interaction of its catalytic core with PABP1 in stress granule formation [36,37]. In the present study, our results revealed that the C-terminal region of DDX3 directly interacts with hnRNPK (Figure 3B), and this interaction is arginine methylation-dependent (Figure 3D). In addition, the deletion of the C-terminal region of DDX3 significantly reduced its protein interaction with hnRNPK^MD^ (Figure 4A) and the apoptosis enhancement mediated by hnRNPK^MD^ in U2OS-2RK cells under DNA damage conditions (Figure 4B,C). Notably, the C-terminal region of DDX3 has been shown to participate in RNA duplex unwinding [33]. In addition, positively charged residues such as Asn^551^, His^578^, and His^579^ in the C-terminus of DDX3 have been reported to be essential to the RNA binding of DDX3 [38]. It is therefore suggested that hnRNPK may interact with the C-terminal region of DDX3 to regulate their molecular interactions in the corresponding regions.

Both DDX3 and hnRNPK function as RNA-binding proteins, and some of the interacting RNAs of DDX3 or hnRNPK may be involved in the apoptosis enhancement promoted by hnRNPK–DDX3 interaction. Both DDX3 and hnRNPK have been shown to regulate the translation of specific genes by binding to their RNAs [6,7,21,22,25,39,40,41,42,43]. Among them, JUND mRNA was previously reported to be stabilized by a protein including hnRNPK and DDX3 [25]. The present study demonstrated that the increased hnRNPK–DDX3 interaction in U2OS cells induced by RK-33 treatment significantly upregulated JUND levels (Figure 6E). Because JUND is positively correlated with apoptosis [44], hnRNPK–DDX3 interaction may promote apoptosis through the stabilization of JUND. Thus, whether downstream pro-oxidant and proinflammatory genes regulated by JUND are also involved in the apoptosis enhancement promoted by hnRNPK–DDX3 interaction should be extensively investigated.

Although both RK-33 and AVN A are small molecule helicase inhibitors of DDX3, only RK-33 could promote hnRNPK–DDX3 interaction as well as apoptosis upon DNA damage (Figure 6 and Appendix A). These results suggest that the docking of different ligands at the ATP binding site of DDX3 may lead to distinct conformations that favor or disfavor hnRNPK–DDX3 interaction. It has been proposed that DDX proteins can switch their conformations from the open form to the closed form through binding with ATP and target RNAs [45,46]. While the crystal structure of ATP/RNA-bound DDX3 is not yet available, the crystal structure of eIF4AIII (DDX48) has been solved and exhibits a transition from open to closed conformation upon binding to AMPPNP and RNA [47]. In contrast to the closed conformation of ATP/RNA-bound eIF4AIII, AMP-bound DDX3 remains in an open conformation [47]. In addition, we further showed that the C-terminal regions of the apo, AMP-, and AMPPNP-bound DDX3 complexes are all different in terms of spatial arrangement (Figure 7D), suggesting that conformational changes may occur in the C-terminal region of DDX3 upon binding with different ATP site-binding ligands. Because RK-33 was designed to dock at the ATP binding sites of DDX3, we hypothesize that RK-33 docking on DDX3 may result in a C-terminal conformation favoring hnRNPK–DDX3 interaction, which is different from the conformation of AMPPNP or AMP-bound DDX3.

Several studies have revealed that RK-33 treatment causes the cell cycle arrest and apoptosis of different cancer cell types, including Ewing sarcoma, lung cancer, colorectal cancer, prostate cancer, and breast cancer cells [30,48,49,50,51]. In addition, RK-33 was shown to serve as a radiation sensitizer that promoted lung tumor regression in a preclinical cancer model [30]. Our study demonstrated that RK-33 is able to promote hnRNPK–DDX3 interactions and is able to enhance apoptosis mediated by hnRNPK^MD^ in U2OS cells under DNA damage conditions. It is possible that DDX3–hnRNPK interaction plays a role in the pharmacological activity of RK-33 and provides new insight into the RK-33-mediated regulation of cancer cells.

In summary, the stronger DDX3–hnRNPK interaction in U2OS-2RK cells versus U2OS-WT cells is crucial to the apoptosis enhancement mediated by hnRNPK^MD^. In addition, treatment with RK-33 in U2OS-2RK cells significantly promotes this apoptosis enhancement. These results suggest that hnRNPK can switch between its apoptosis-suppressive and apoptosis-promoting roles through the methylation-dependent DDX3–hnRNPK interaction. It is therefore important to closely monitor the arginine methylation level of hnRNPK in U2OS cells, which is strongly associated with the capacity to induce DDX3–hnRNPK interaction. On the other hand, RK-33 treatment provides a feasible strategy for selectively promoting the DDX3 interaction with unmethylated hnRNPK in cancer cells. As shown in Appendix A, RK-33 treatment in original U2OS osteosarcoma cells under DNA damage conditions could also increase apoptosis. Therefore, our discovery of the proapoptotic role of the DDX3–hnRNPK interaction in U2OS cells may facilitate the development of an apoptosis-promoting strategy against cancer development.

## Figures and Tables

**Figure 1 ijms-22-09764-f001:**
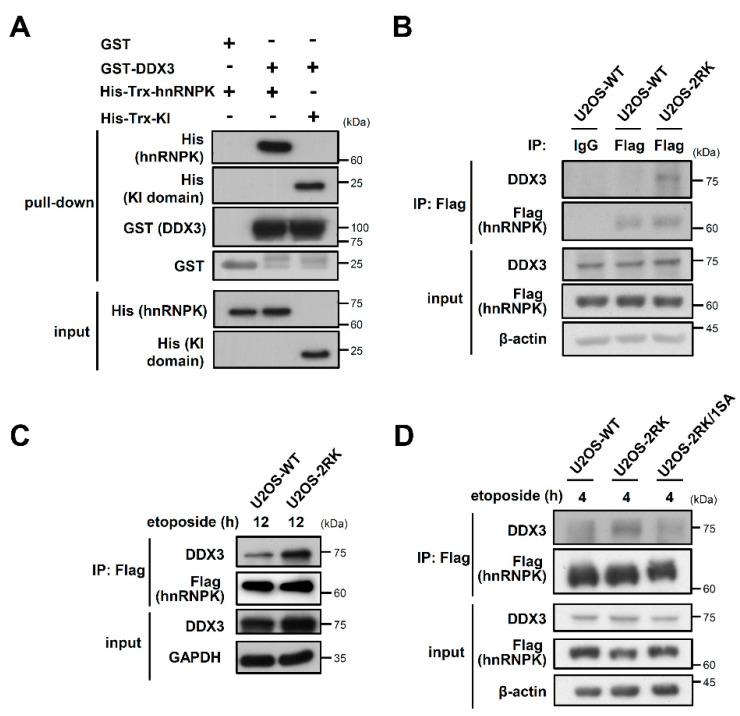
Loss of Arg296/299 methylation in hnRNPK promotes the DDX3–hnRNPK interaction in vivo upon DNA damage. (**A**) Recombinant hnRNPK or Trx-KI was incubated with GST-DDX3, followed by SDS-PAGE to determine the interaction between hnRNPK or KI and the DDX3 proteins. (**B**) Flag-tagged hnRNPK was precipitated from either U2OS-WT or U2OS-2RK cells using a Flag-specific antibody. Subsequent measurement of the precipitated DDX3 by hnRNPK was performed using a DDX3-specific antibody. (**C**) U2OS-WT cells and U2OS-2RK cells were first treated with etoposide (50 μM), respectively, followed by the immunoprecipitation of Flag-hnRNPK. Subsequent measurement of the precipitated DDX3 was performed using DDX3-specific antibodies. (**D**) U2OS-WT, U2OS-2RK, and U2OS-2RK/1SA cells were first treated with etoposide (50 μM) followed by the immunoprecipitation of Flag-hnRNPK. Subsequent measurement of the precipitated DDX3 by either wild-type hnRNPK or hnRNPK^MD^ with or without the S302A mutation was performed using DDX3-specific antibodies.

**Figure 2 ijms-22-09764-f002:**
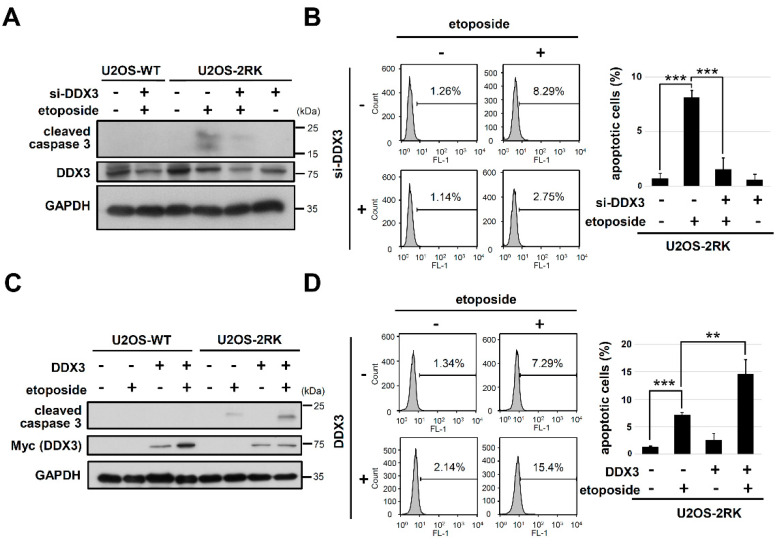
Knockdown of DDX3 attenuates hnRNPK^MD^-mediated apoptosis, whereas the overexpression of DDX3 promotes the induction of this apoptosis. (**A**) U2OS-WT and U2OS-2RK cells were transfected with or without DDX3 siRNA and were incubated for 24 h followed by etoposide (50 μM) treatment for another 12 h. Cells were collected and were analyzed for the expression levels of cleaved caspase-3, DDX3, and GAPDH using specific antibodies. (**B**) Under the same treatment as described above, U2OS-2RK cells were collected and analyzed by TUNEL assay to determine the percentage of apoptotic cells using FACS (left). Quantitation of apoptotic cells in the TUNEL assay is shown (right). (**C**) U2OS-WT and U2OS-2RK cells were transfected with either pcDNA4 vector or Myc-DDX3 and were incubated for 24 h followed by treatment with etoposide (50 μM) for another 12 h. Cell lysates were collected and were assessed by Western blot analysis to determine the expression levels of cleaved caspase-3, DDX3, and GAPDH. (**D**) Under the same treatment as described above, U2OS-2RK cells were collected and analyzed by TUNEL assay to determine the percentage of apoptotic cells using FACS (left). Quantitation of apoptotic cells in the TUNEL assay is shown (right). ** as *p <* 0.01, *** as *p <* 0.001.

**Figure 3 ijms-22-09764-f003:**
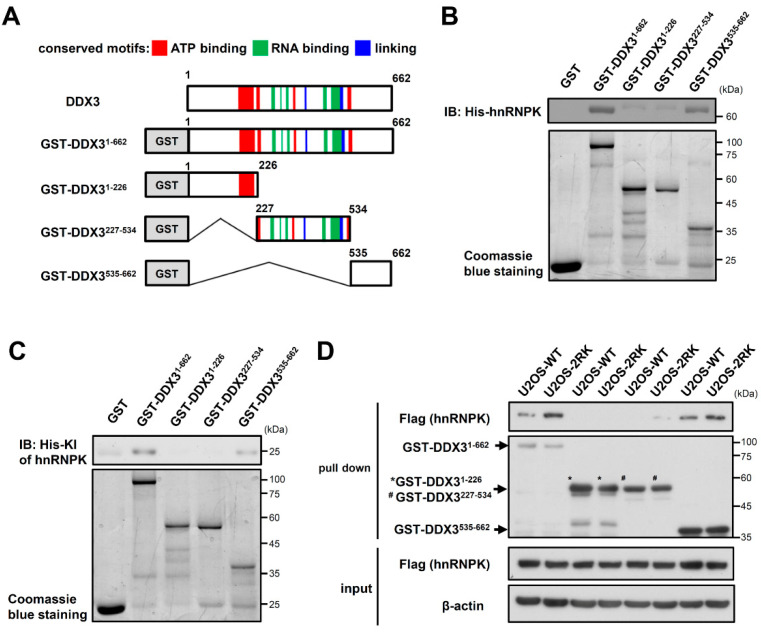
The C-terminal region of DDX3 is responsible for hnRNPK–DDX3 interaction. (**A**) Schematic representation of GST-DDX3 variants established in this study. The red box indicates ATP-binding motifs, the green box indicates RNA-binding motifs, and the blue box indicates the linking area between the ATP and RNA binding sites. Pull-down assays were conducted by incubating each GST-DDX3 variant with either recombinant his-hnRNPK (**B**) or his-KI (**C**), followed by SDS-PAGE to determine the interaction between hnRNPK or the KI and DDX3 variants. (**D**) U2OS-WT or U2OS-2RK cell lysates were incubated with GST-DDX3 variants in pull-down assays to determine whether the methylation of hnRNPK affects its interaction with diverse DDX3 variants. * as GST-DDX^1–226^, **^#^** as GST-DDX3^227^^–^^534^.

**Figure 4 ijms-22-09764-f004:**
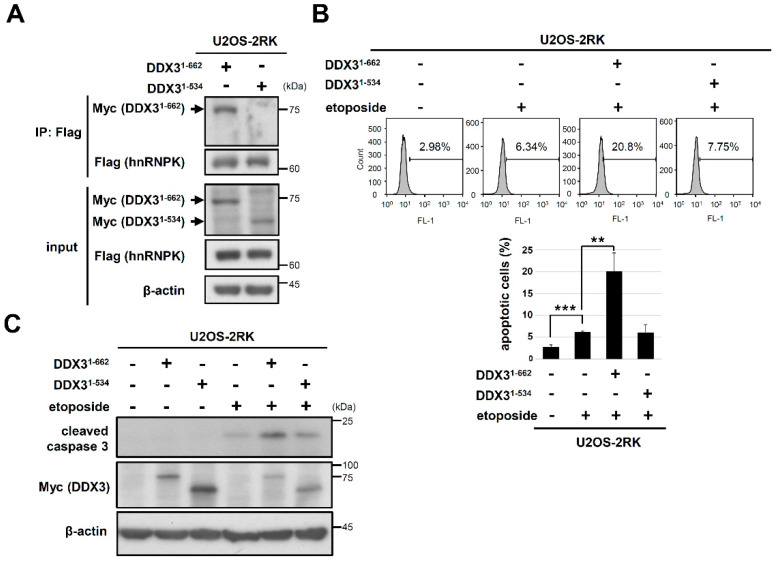
Truncation of the C-terminus in DDX3 diminished hnRNPK^MD^-mediated apoptosis upon DNA damage. (**A**) U2OS-2RK cells were transfected with either Myc-DDX3^1–662^ or Myc-DDX3^1–534^ for 48 h. Cell lysates were collected to measure the interaction between hnRNPK and DDX3. (**B**) U2OS-2RK cells were transfected with either pcDNA4, Myc-DDX3^1–662^, or Myc-DDX3^1–534^ for 24 h and were then treated with etoposide (50 μM) for another 12 h. Cell lysates were collected and analyzed by TUNEL assay to determine the degree of apoptosis using FACS (top). Quantitation of apoptotic cells in the TUNEL assay is shown (bottom). (**C**) Under the same treatment as described above, cell lysates were collected and analyzed for the expression levels of active caspase-3. ** as *p <* 0.01, *** as *p <* 0.001.

**Figure 5 ijms-22-09764-f005:**
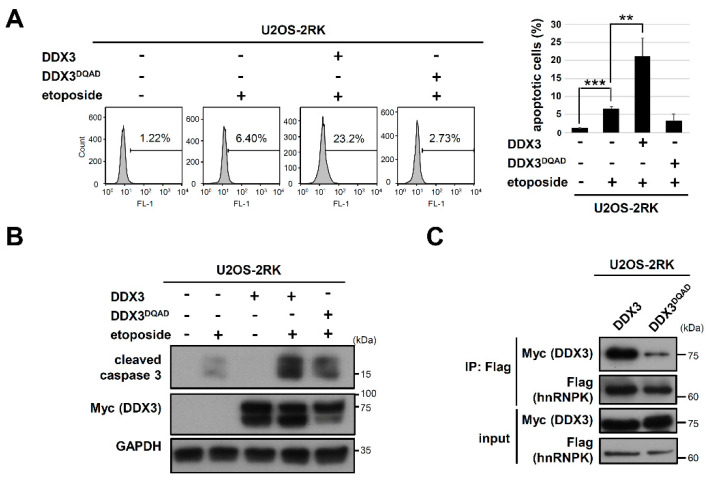
The helicase-inactive DDX3^DQAD^ reduces hnRNPK^MD^-mediated apoptosis by impairing hnRNPK–DDX3 interaction. (**A**) U2OS-2RK cells were transfected with either pcDNA4, Myc-DDX3, or Myc-DDX3^DQAD^ for 24 h followed by etoposide (50 μM) treatment for another 12 h. Cell lysates were collected and analyzed by TUNEL assay to determine the degree of apoptosis using FACS (left). Quantitation of apoptotic cells in the TUNEL assay is shown (right). (**B**) Under the same treatment as described above, cell lysates were collected and analyzed by Western blot to determine the expression levels of cleaved caspase-3, DDX3, and GAPDH. (**C**) U2OS-2RK cells were, respectively transfected with either Myc-DDX3 or Myc-DDX3^DQAD^ for 36 h followed by immunoprecipitation of hnRNPK-2RK. Subsequent detection of the precipitated Myc-DDX3 or Myc-DDX3^DQAD^ by hnRNPK was performed. ** as *p <* 0.01, *** as *p <* 0.001.

**Figure 6 ijms-22-09764-f006:**
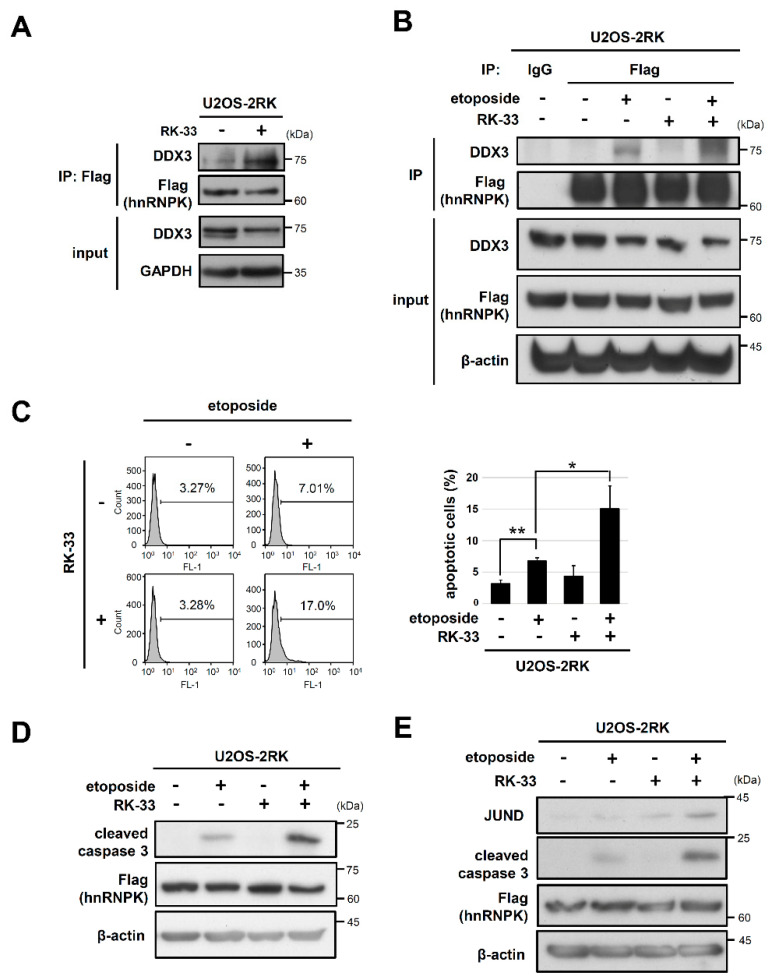
A DDX3 inhibitor, RK-33, enhances hnRNPK^MD^-mediated apoptosis by increasing hnRNPK–DDX3 interaction. (**A**) U2OS-2RK cells were treated with RK-33 (10 μM) for 12 h. Cell lysates were collected to determine the interaction between hnRNPK and DDX3. (**B**) U2OS-2RK cells were treated with etoposide only (50 μM), RK-33 only (5 μM), or etoposide/RK-33 together for 4 h. Cell lysates were collected to determine the interaction between hnRNPK and DDX3. (**C**) Under the same treatment as described above, cells were collected at 12 h and were analyzed by TUNEL assay to determine the degree of apoptosis using FACS (left). Quantitation of apoptotic cells in the TUNEL assay is shown (right). (**D**) U2OS-2RK cells were treated with etoposide only (50 μM), RK-33 only (5 μM), or etoposide/RK-33 together for 12 h. The treated cells were collected, and the protein levels of active caspase-3 and (**E**) JUND were assessed. * as *p <* 0.05, ** as *p <* 0.01.

**Figure 7 ijms-22-09764-f007:**
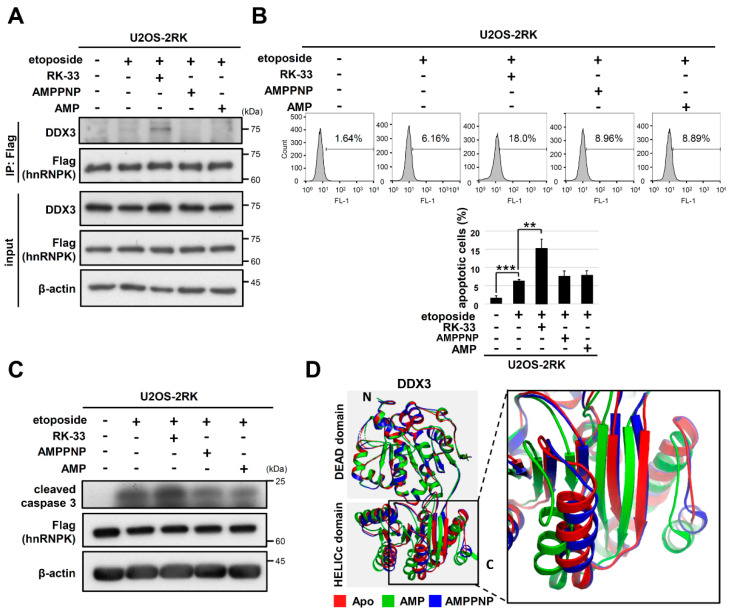
Treatment with AMPPNP and AMP exhibited less enhancement of hnRNPK^MD^-mediated apoptosis than RK-33 treatment. (**A**) U2OS-2RK cells were treated with etoposide only (50 μM), etoposide together with RK-33 (5 μM), AMPPNP (5 μM), or AMP (5 μM) for 4 h. Cell lysates were collected to determine the interaction between hnRNPK and DDX3. (**B**) Under the same treatment as described above, cell lysates were collected at 12 h and were analyzed by TUNEL assay to determine the degree of apoptosis using FACS (top). Quantitation of apoptotic cells in the TUNEL assay is shown (bottom). (**C**) Under the same treatment as described above, cell lysates were collected at 12 h. Cell lysates were collected and analyzed for the expression levels of active caspase-3. (**D**) Structural comparison among the apo form of hDDX3 (PDB: 5e7i) and hDDX3 cocrystallized with either AMPPNP (PDB: 5e7j) or AMP (PDB: 5e7m) was conducted. The zoomed-in square indicates the major difference in the C-terminal conformation between AMPPNP-bound hDDX3 and AMP-bound hDDX3. ** as *p <* 0.01, *** as *p <* 0.001.

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
