# Peer review of "Arginine Methylation of hnRNPK Inhibits the DDX3-hnRNPK Interaction to Play an Anti-Apoptosis Role in Osteosarcoma Cells"

_ijms, 2021, doi:10.3390/ijms22189764_

Round 1
Reviewer 1 Report
- The article is well represented by the authors.
- The aim of the work is well developed in the description of the results and the conclusion.
- The scientific literature used could be updated.
Author Response
Reviewer 1
Point 1: The article is well represented by the authors.
Response:
The authors thank the Reviewer's comment.
Point 2: The aim of the work is well developed in the description of the results and the conclusion.
Response:
The authors thank the Reviewer's comment.
Point 3: The scientific literature used could be updated.
Response:
The authors thank the Reviewer's suggestion and have accordingly either replaced the relatively old references with the more recent studies or added the updated study. In addition, these references were highlighted in red color in the revised manuscript. In details:
- Introduction
Ref. 7 The old reference (Bomsztyk et al., 2004) was replaced by a recent study (Wang et al., 2020) for the introduction of hnRNPK’s function.
Ref. 12, 13 Two old studies (Chen et al., 2008; Matta et al., 2009) were replaced by the updated reports (Barboro et al., 2014; Yang et al., 2016) to illustrate the association of hnRNPK with poor prognosis in cancers.
Ref 14 The old study (Moumen et al., 2005) was replaced by the updated report (Lee et al., 2012) to describe the relation between hnRNPK and p53 in cell cycle arrest and apoptosis upon DNA damage.
Ref 19 The wrong citation was replaced by the corrected study (Mikula et al., 2006) to implicate the possible presence of DDX3 and hnRNPK in the same protein complex.
- Results
Ref. 29 In addition to the original study, an updated report (de Bisschop et al., 2019) was also added as a reference to further explain the influence of DQAD mutation on the RNA helicase activity of DDX3.
Ref. 33 The old reference (Li et al., 2002) was replaced by a updated study (Papoudou-Bai et al., 2017) to illustrate the association between JUND and apoptosis.
- Discussion
Ref. 42 The old reference (Notari et al., 2006) was replaced by the recent study (Gallardo et al., 2020) to correlate with the translational regulation of Myc RNA by hnRNPK

Reviewer 2 Report
In line with the results from a previous paper, authors here demonstrate that the stronger interaction between DDX3 and hnRNPK in U2OS-2RK, in comparison with U2OS-WT, plays a key role in the enhancement of the apoptosis mediated by hnRNPKMD.
The paper is well-written ad data are clearly presented and discussed.
The introduction covers the major points related to the scientific background of the research and the methodologies are up-to-date and correctly employed.
Author Response
The authors appreciate the Reviewer's comment. Thank you.
